# Bio-Inspired Hierarchical Micro-/Nanostructures for Anti-Icing Solely Fabricated by Metal-Assisted Chemical Etching

**DOI:** 10.3390/mi13071077

**Published:** 2022-07-07

**Authors:** Lansheng Zhang, Xiaoyang Chu, Feng Tian, Yang Xu, Huan Hu

**Affiliations:** 1ZJU-UIUC Institute, International Campus, Zhejiang University, Haining 314400, China; lansheng.20@intl.zju.edu.cn (L.Z.); xiaoyang.20@intl.zju.edu.cn (X.C.); tianfeng@zju.edu.cn (F.T.); 2School of Micro-Nano Electronics, Zhejiang University, Hangzhou 310000, China; yangxu-isee@zju.edu.cn; 3State Key Laboratory of Fluid Power and Mechatronic Systems, Zhejiang University, Hangzhou 310000, China

**Keywords:** hierarchical micro-/nanostructures, superhydrophobic, anti-icing, metal-assisted chemical etching

## Abstract

We report a cost-effective and scalable methodology for producing a hierarchical micro-/nanostructured silicon surface solely by metal-assisted chemical etching. It involves two major processing steps of fabricating micropillars and nanowires separately. The process of producing micro-scale structures by masked metal-assisted chemical etching was optimized. Silicon nanowires were created on the micropillar’s surface via maskless metal-assisted chemical etching. The hierarchical micro-/nanostructured surface exhibits superhydrophobic properties with a high contact angle of ~156° and a low sliding angle of <2.5° for deionized water. Furthermore, due to the existence of microscale and nanoscale air trapped at the liquid/solid interface, it exhibits a long ice delay time of 2876 s at −5 °C, more than 5 times longer than that of smooth surfaces. Compared to conventional dry etching methods, the metal-assisted chemical etching approach excludes vacuum environments and high-temperature processes and can be applied for applications requiring hierarchical micro-/nanostructured surfaces or structures.

## 1. Introduction

Nature provides a variety of functional surfaces, whose excellent properties provide inspiration to develop new techniques and methodologies to construct advanced artificial materials. Superhydrophobic surfaces, with high water contact angles (>150°) and low sliding angles (<10°), have received tremendous attention in recent years. Superhydrophobicity is generally obtained by surface texture or chemical properties [1,2,3] and the fabrication methods are commonly categorized into two types: the top-down approach and the bottom-up approach [4]. Many superhydrophobic surfaces have been applied in diverse fields, such as space and aerospace, defense, automotive, biomedical applications, sensors, apparel, and so on [5,6,7]. Further, for different application environments, various methods have been proposed to improve durability, corrosion resistance, chemical, and mechanical stability [8,9,10]. Considerable progress in fabrication, characterization, and applications of superhydrophobic surfaces has been obtained. However, there are also some challenges for further investigation, such as surface robustness, optical transparency, cost issues, etc. [4,11,12].

Many biological superhydrophobic surfaces such as lotus leaves [13,14,15], red rose petals [16], and butterfly wings [17] exhibit hierarchical micro-/nanostructures. In addition, many other biological components including compound lens [18], gecko feet [19], and even hairs all exhibit hierarchical micro-/nanostructures crucial for achieving superior functions. The hierarchical micro-/nanostructure is crucial to the superhydrophobic property because the composite surface consists of solid surfaces and air pockets, which increases the contact angle of water [20,21]. Currently, hierarchical micro-/nanostructures consisting of different materials such as Si [22], SiO_2_ [23], TiO_2_ [24], SnO_2_ [25], MgO [26] are being demonstrated and applied in superhydrophobic [27], self-cleaning [28], fog collection [29], oil-water separation [30], anti-icing [31,32], energy [33,34,35,36], and sensing [37] applications.

The reported fabrication methods of hierarchical micro-/nanostructures (MN) include nanosecond pulsed laser irradiation [38], femtosecond laser [39], nanoimprinting [40], hot-embossing [41], scanning probe lithography [42], glancing angle deposition [43], self-assembly [44,45], metal-assisted chemical etching [46], etc. The traditional methods for forming hierarchical micro-/nanostructures are shown in Appendix A. Most of these methods rely on costly equipment [38,39,42,47] and require a high-temperature process [41] or vacuum environment [43]. Moreover, many of them require two different fabrication processes [39] or materials [43] to form the hierarchical micro-/nanostructures.

In this paper, we report a new process of producing a hierarchical micro-/nanostructured (MN) surface solely by metal-assisted chemical etching (MacEtch). The first step is to produce micropillars using masked MacEtch with micropatterns defined by conventional ultraviolet lithography. The etching conditions are optimized to minimize metal film detachment and over-etching of top parts of micropillars to render high-quality silicon micropillars. The second step is to produce silicon nanowires via maskless MacEtch. After the hierarchical MN surfaces are produced, a fluorocarbon film was deposited for hydrophobicity. The wettability characterization results show that the hierarchical MN surface exhibits a large contact angle of 156° and a small sliding angle of less than 2.5°. Moreover, a long icing delay time (IDT) of 2876 s is demonstrated for the hierarchical MN surface that is 5 times longer than the IDT on a smooth surface at −5 °C temperature. Our fabrication approach solely based on MacEtch is cost effective and scalable. Further, the metal-assisted chemical etching excludes the use of vacuum environment and high-temperature processes and can be applied for applications requiring hierarchical MN surfaces or structures.

## 2. Experiment

### 2.1. Fabrication Process

The substrate used was an *n*-type <100> silicon wafer with an electrical resistivity of 1~10 Ω·cm obtained from Suzhou Research Materials Microtech Co., Ltd., Suzhou, China. The photoresist used for conventional UV photolithography was the positive photoresist AR-P 5350 obtained from GermanTech Co., Ltd, Strausberg, Germany. The source materials for thermal evaporation were gold pellets (Au 13,503, V: 3 mm × 3 mm, purity: 99.999%) obtained from Zhong Nuo Advanced Material Technology Co., Ltd., Beijing, China. The silver nitrate particles and liquid chemicals of the hydrofluoric acid, hydrochloric acid, nitric acid, hydrogen peroxide, and ammonium hydroxide were obtained from Sinopharm Chemical Reagent Co., Ltd., Shanghai, China.

Figure 1 shows the major fabrication steps of the hierarchical MN surface solely using MacEtch. First, the photoresist was patterned via conventional UV photolithography. Then a 13-nm-thick gold film was deposited via thermal evaporation and a lift-off process was applied to produce the gold patterns (step a). Second, the masked MacEtch process was employed to fabricate the micropillars (step b), after which the gold film was removed by chemical etching. Third, the wafer underwent ultrasonic treatment and thermal oxidation at 1100 °C to form a 2000 Å silicon oxide layer which later was etched by hydrofluoric acids. This step was aimed at smoothing the rough sidewall surfaces of micropillars during the masked MacEtch process. Fourth, silicon nanowires were produced on the micropillars’ surface both on top and on sidewalls by maskless MacEtch using HF/AgNO_3_ solution (step c). Finally, an amorphous fluorocarbon film was deposited on the surface to reduce the surface energy and improve water resistance (step d).

### 2.2. Metal-Assisted Chemical Etching Process for Hierarchical Micro-/Nanostructured Surface

#### 2.2.1. Metal-Assisted Chemical Etching Process with Gold

Figure 2 illustrates the mechanism of MacEtch with patterned gold according to published reports [48]. First, (1) the oxidant (H_2_O_2_) is preferentially reduced on the silicon surface covered with the catalyst (gold), and the holes generated by the reduction reaction of the oxidant diffuse through the gold and are injected into the silicon beneath the gold film. (2) Then, the Si is oxidized by the injected holes and dissolved at the Si/Au interface by HF. The byproducts diffuse along the interface between the Si and the gold film. (3) The holes diffuse from the Si beneath the gold film to off-metal areas if the rate of hole consumption at the Si/Au interface is smaller than the rate of hole injection. However, the concentration of holes at the Si/Au interface is much higher than off-metal areas, so the silicon at the Si/Au interface would be etched much faster than off-metal areas.

#### 2.2.2. Metal-Assisted Chemical Etching for Micropillars

The first step in forming the hierarchical MN surface is to produce the micropillars by masked MacEtch. In this step, the thickness of the gold film, as well as the mixing ratio of the etchants, are two crucial parameters. Although the gold film is chemically stable in etchants, the limited adhesion between the gold film and the silicon surface cannot hold the integrity and assure adhesion in non-ideal etching conditions. Although increasing the thickness of the metal film can promote cohesion, the film becomes dense and limits the lateral diffusion of the etchant/by-product, resulting in an extremely slow etching rate, and even failures of etching. The gold, on the other hand, becomes a continuous film at ≈10 nm or higher [49]. We chose a 13 nm gold film, which was deposited by thermal evaporation at a deposition rate between 0.4 Å/s–0.6 Å/s.

The mixing ratio of two etchants HF and H_2_O_2_ plays an important role in the final morphology of etched structures. Therefore, we performed an optimization of the mixing ratio experimentally to identify the optimal mixing ratio. Figure 3 depicts the micropillar etching results at different mixing ratios denoted by *ρ*, wherein *ρ* = [HF]/[HF] + [H_2_O_2_], [HF], and [H_2_O_2_] are respectively the molar concentrations of HF and H_2_O_2_ with a unit of moles per liter [H_2_O_2_] = 2.26 M. The results show that the etching process almost stops at *ρ* ≤ 0.2 (Figure 3a) because of the low concentration of HF cannot consume the oxidized silicon. As the concentration of HF increased to *ρ* = 0.3 (Figure 3b), the formed micropillars were inclined and the upper surface was severely etched due to the diffusion of excess holes (Figure 2, process 3). Figure 3c shows the formed micropillars and other small micro-/nanostructures on etched surfaces. The generation of these small micro-/nanostructures is a nuisance, and will be optimized later. Although increasing the concentration of HF can further increase the etching rate and reduce the attack on the top and sidewall surfaces of the micropillars, the higher etching rate can lead to delamination of the gold metal mesh film [49], as shown in Figure 3d. The movement and delamination of gold film render deformed micropillars and are not desired in our application. Therefore, we chose *ρ* = 0.6 for the masked MacEtch step.

Figure 4a shows micropillars produced by MacEtch (*ρ* = 0.6, [H_2_O_2_] = 2.26 M, t = 28 min). There are a small number of undesired smaller micro-/nanostructures due to defects or pores in the gold film. To remove them to ensure good quality, we implemented ultrasonic treatment to physically remove them. In addition, we employed thermal oxidation and HF wet etching to chemically remove those small micro-/nanostructures. This oxidation and HF etching were efficient in smoothening the top surface as well as the sidewall of silicon micropillars to promote the formation of nanowires in the next stage by MacEtch [50]. Figure 4c shows the silicon micropillars after both physical and chemical treatment aforementioned. After the ultrasonic treatment (35 W for 4 min) and surface oxidation (200 nm thickness and dissolved in HF), undesired micro-/nanostructures were successfully removed. The oxidation process also smooths the micropillar surface, as shown in Figure 4b,d.

#### 2.2.3. Maskless MacEtch for Producing Silicon Nanowires

The second step in producing the hierarchical MN is to fabricate the nanowires on micropillars. Before this step, the gold film used in the first stage was etched and removed using the etchant (HCL (38%): HNO_3_ (68%) 3:1). The silicon nanowires were produced by electroless etching in a solution of AgNO_3_ (20 mM) in HF (5 M) for about 2 min. This method depends on the two-step kinetics of the Ag^+^−Si redox couple [51]. The Si was oxidized on the surface by the holes induced by the reduction of Ag^+^; then, the oxide was etched away by HF to form an ordered self-assembly of nanopillars. After the synthesis of the nanowires, the sample was cleaned in the solution (H_2_O_2_:NH_4_OH = 3:1) to etch away silver dendrites deposit and reveal silicon nanopillars. Figure 5a shows the hierarchical MN surface. The nanowires are mainly distributed on the top surface and the upper side wall of the micropillars, as shown in Figure 5b. Other areas of the micropillars’ sidewall are nanopores, and the surface among the micropillars does not have nanowires either. This can be attributed to the concentration gradients of etchants that lower concentration of HF can result in porous morphology other than nanopillar morphology [52]. Appendix A shows the measured statistic distribution of the nanowire height mainly ranging between 80 nm and 200 nm. Figure 5c,d show the distribution of nanowires on the top surface and side wall of the micropillar respectively. These nanowires further reduce the contact area between solid/liquid and increase the interface of air/liquid, and therefore are crucial to improving the superhydrophobicity and water-repelling capability [32,53].

### 2.3. Wettability Measurements

The well-known Cassie–Baxter theory is widely used to characterize the wetting behavior of a surface. The equilibrium contact angle (CA) of a composite surface where vapor pockets are trapped underneath the liquid is expressed by the following Equation (1) [50]:(1)$$cos\theta^{*}=f\left(cos\theta_{y}+1\right)-1\;$$
where $\theta^{*}$ is the apparent CA, f is the area fraction of the solid that is in contact with the liquid, and $\theta_{y}\;$ is the equilibrium CA of the liquid droplet on a smooth surface. Equation (1) shows that the f and $\theta_{y}$ can be adjusted to increase the CA. f can be reduced by increasing the surface roughness, and $\theta_{y}$ is increased by covering low-surface-energy materials. In this paper, we fabricated the hierarchical micro-/nanostructure, microstructure, and nanostructure on three different surfaces to reduce the area fraction, and added the low-surface-energy material fluorocarbon to increase the CA of the smooth surface.

Four different surfaces, hierarchical micro-/nanostructured surface (MN), microstructured surface (M), nanostructured surface (N), and smooth surface (S) were designed to characterize the contributions of different surfaces to superhydrophobicity. All the surfaces were coated with an amorphous fluorocarbon film. The water droplet image was taken using a contact angle measuring system, and the water contact angle was measured using Image J software, version 1.53e, accessed on 2 June 2022. Three duplicate measurements were taken under normal ambient laboratory conditions.

### 2.4. Anti-Icing Property Measurements

Anti-icing ability measurements were performed with the aim of characterizing the icing delay time (IDT) of the four different surfaces. Appendix A shows the measurement platform, which was designed using a Peltier thermoelectric generator sandwiched between a copper plate and a water-cooling unit. A digital temperature controller was attached to the platform to regulate the temperature. A hygrometer was used to detect ambient humidity. A 7 μL water droplet was used on a 1 cm × 1 cm sample area, and the time taken for the water droplet to turn into ice was recorded. Three duplicate measurements were taken for all the samples at ambient conditions (25 °C temperature and 21–26% relative humidity).

## 3. Results and Discussion

### 3.1. Wettability Analysis

Figure 6 shows the equilibrium water CAs of S, N, M, and MN surfaces, which are plotted on the theoretical Cassie state curve. The equilibrium water CA of the S surface with fluorocarbon film is 108° and the area fraction (f) is ~1. The equilibrium water CA of the N surface is 132°, and the f value obtained from Equation (1) is 0.48. The equilibrium CA of the M surface is 147°, corresponding to an f value of 0.23, and the f value obtained in Figure 4c is 0.2, which indicates a good agreement between the experimental and theoretical results. For the MN surface, the equilibrium CA is 156° and the f value is 0.12, which is close to the theoretical f value of 0.11. All these results show that the water droplet had a Wenzel wetting state contact with the S surface, but showed a Cassie state contact with the N, M, and MN surfaces. This can be attributed to the composite nature of the M and MN surfaces made of solid materials and trapped air. Additionally, the highest CA value of the MN surface is due to the hierarchical nature of the surface. These findings also prove that superhydrophobicity is largely dependent on multi-scale structures. Furthermore, Appendix A demonstrates that water droplets can easily roll off the MN surface at a very low tilt angle (<2.5°). This attribute is essential for self-cleaning applications.

To further increase the contact angle of the hierarchical micro-/nanostructured surface, both microstructures and nanostructures can be optimized. The micropillar gap, area fractions, and heights can be easily defined by the UV-lithography and the MacEtch process with gold. Further, the higher concentration of H_2_O_2_ in etchant (Ag+ and HF) can also produce lower-density nanowire arrays [54]. Both of these methods can improve the contact angle of the hierarchical micro-/nanostructured surface.

### 3.2. Anti-Icing Analysis

To characterize the anti-icing properties of the MN surfaces, we placed water droplets on samples atop temperature-controllable stages and recorded the duration (IDT) needed for droplets to turn from water to ice. Figure 7 shows the measurement results of icing delay time (IDT) of 7 μL water droplets on the S, N, M, and MN surfaces. The images show the gradual transformation of the droplet from liquid to ice. The droplet on the S surface shows the least IDT of ~488 s at −5 °C, suggesting that a normal smooth silicon surface has weak anti-icing properties. The icing delay time for the droplet on the N surface and the M surface were 1982 s and 2143 s respectively. The hierarchical MN surface offered the highest IDT of 2876 s at −5 °C, which is 5 times the IDT of the S surface, indicating that it offers significantly improved anti-icing properties compared to the other structured surfaces. Appendix A comprises a recording of the whole process of water droplet transition from the liquid state to the ice state for the S and MN surfaces.

Compared to other structured surfaces, the high icing delay time value obtained from the MN surface can be attributed to the fact that heat transfer proceeds largely across the contact area between the water droplets and the structured pillars. Since the water droplet sits atop the air pockets on the MN surface, and due to the low thermal conductivity of air, the MN pillars served as the primary heat conduct path in the vertical direction between the cold silicon substrate and the water droplet, as shown in Figure 8. The air pockets at the solid–liquid interface decrease the heat transfer efficiency. Furthermore, the classical nucleation theory suggests that the larger the CA of the substrate, the greater the free-energy barrier required for the ice nucleus formation, the smaller the rate of nucleation thus slower ice formation [55,56]. The hierarchical micro-/nanostructured silicon surfaces are useful in some silicon-based surface devices such as the anti-icing application of silicon solar cells [57,58].

## 4. Conclusions

A new cost-effective and scalable methodology to produce a hierarchical micro-/nanostructured surface sorely by metal-assisted chemical etching was demonstrated. The advantage of this methodology lies in the fact that all the etching was accomplished by metal-assisted chemical etching without using sophisticated equipment, vacuum environment, and high temperature. The produced hierarchical micro-/nanostructured surface offered a large water contact angle of ~156° and a small sliding angle of <2.5° indicating outstanding self-cleaning potentials. Additionally, the surface showed an excellent icing delay time of 2876 s at −5 °C, more than 5 times the icing delay time of smooth surfaces.

## Figures and Tables

**Figure 1 micromachines-13-01077-f001:**
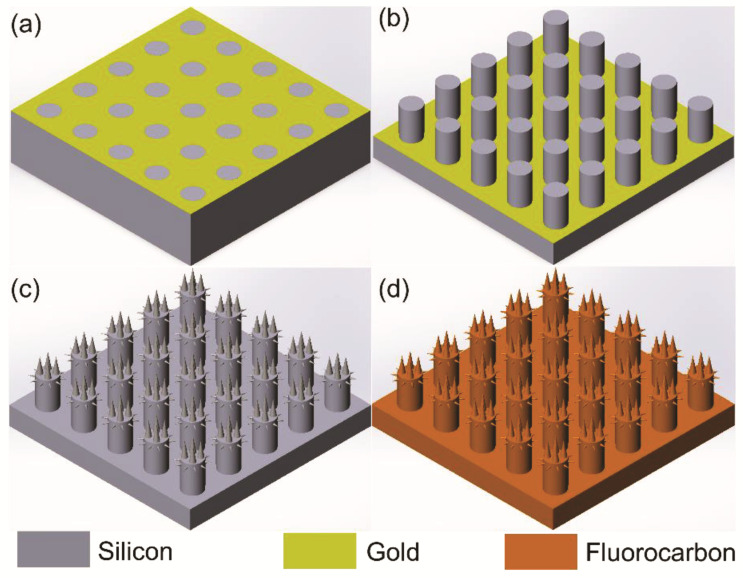
Schematic illustration of the major fabrication processes of hierarchical micro-/nanostructured surface. (**a**) UV photolithography, gold deposition, and lift-off process; (**b**) metal-assisted chemical etching for micropillars; (**c**) metal-assisted chemical etching for nanowires; (**d**) fluorocarbon deposition.

**Figure 2 micromachines-13-01077-f002:**
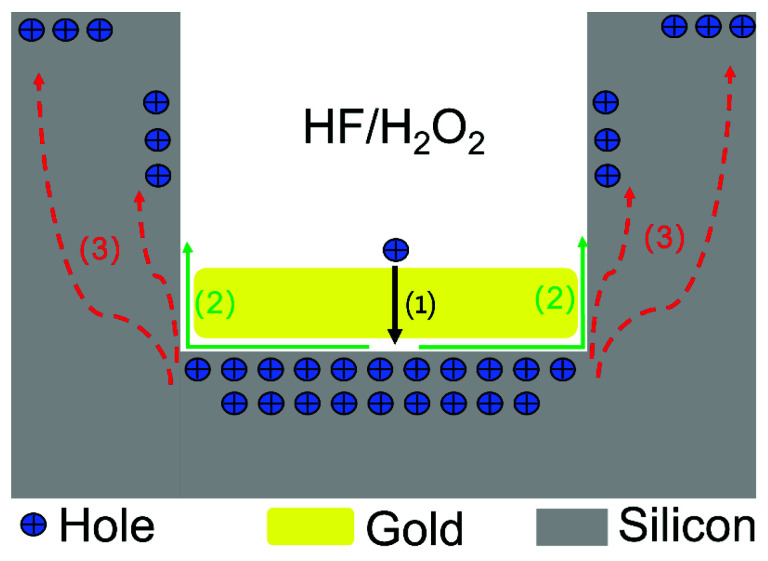
Schematic processes of metal-assisted chemical etching with gold. (1) The oxidant (H_2_O_2_) is reduced, and the holes generate and inject into the silicon beneath the gold film. (2) The Si is oxidized by the injected holes and dissolved at the Si/Au interface by HF. (3) The off-metal areas are etched because of the excess hole diffusion.

**Figure 3 micromachines-13-01077-f003:**
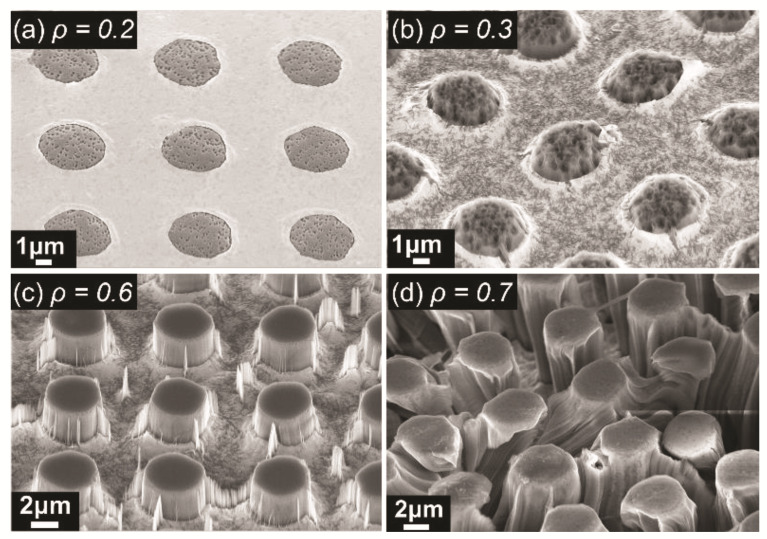
SEM 45° view of micropillars produced by metal assisted chemical etching in etchant with different *ρ*, (**a**) *ρ* = 0.2, (**b**) *ρ* = 0.3, (**c**) *ρ* = 0.6, (**d**) *ρ* = 0.7. (*ρ* = [HF]/[HF] + [H_2_O_2_], [HF] and [H_2_O_2_] are respectively the molar concentration of HF and H_2_O_2_ in moles per liter; [H_2_O_2_] = 2.26 M, t = 20 min).

**Figure 4 micromachines-13-01077-f004:**
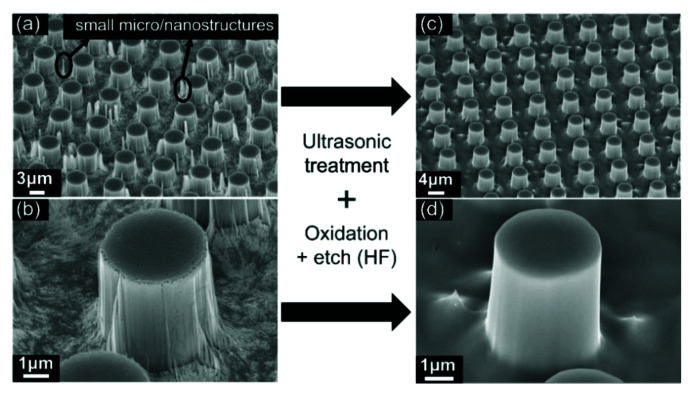
SEM pictures of micropillars produced by metal-assisted chemical etching. (**a**) There are both micropillars and some redundant small micro-/nanostructures on the surface (*ρ* = 0.6, [H_2_O_2_] = 2.26 M, t = 28 min), (**c**) small micro-/nanostructures were almost removed by the process of ultrasonic treatment, surface oxidation and dissolving in HF. The rough surface of micropillars (**b**) became smooth (**d**) by surface oxidation and dissolving in the HF process.

**Figure 5 micromachines-13-01077-f005:**
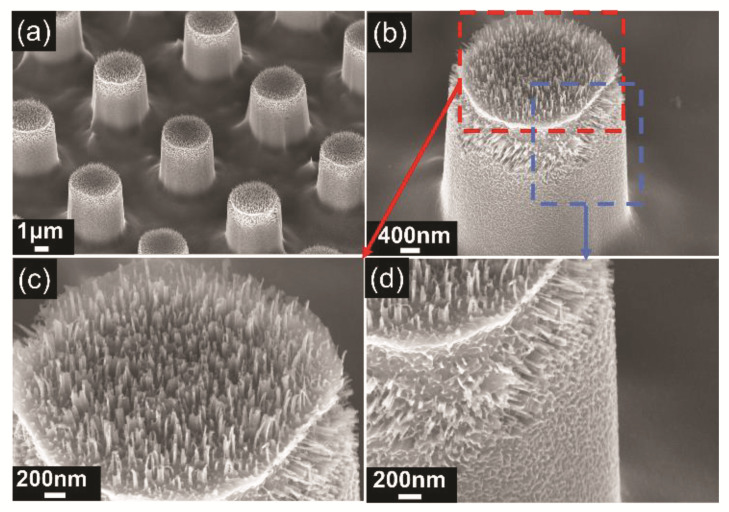
SEM image of the hierarchical micro-/nanostructured surface. (**a**) The hierarchical micro-/nanostructured surface. (**b**–**d**) The distribution of nanowires on the top surface and side wall of the micropillar.

**Figure 6 micromachines-13-01077-f006:**
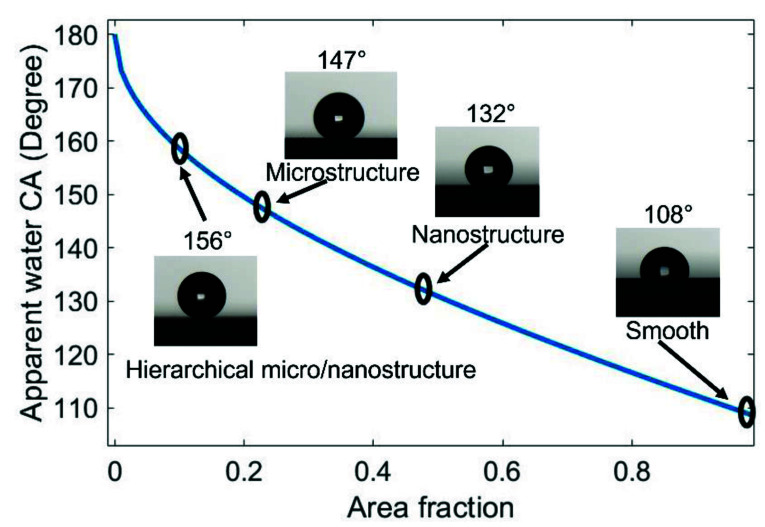
The equilibrium contact angle of waterdrop on four different surfaces compared with the theoretical Cassie state curve.

**Figure 7 micromachines-13-01077-f007:**
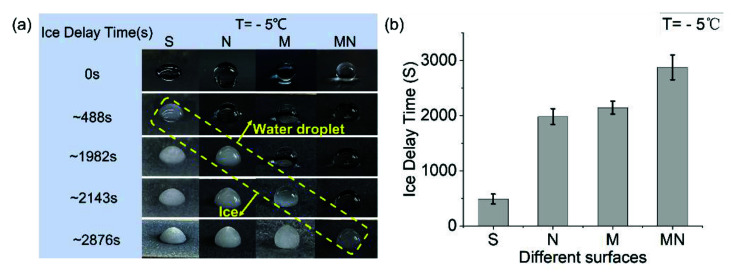
(**a**) Observations of ice formation on different surfaces at −5 °C. From left to right are the photos of the droplets on the surfaces of S, N, M, and MN, respectively. First, the droplet on the S surface freezes, with an IDT of ~488 s; the droplets on M and N surfaces freeze at ~1982 s and 2143 s; and the droplet on MN freezes at ~2876 s. (**b**) Comparison of the icing delay times of water droplets on four different surfaces.

**Figure 8 micromachines-13-01077-f008:**
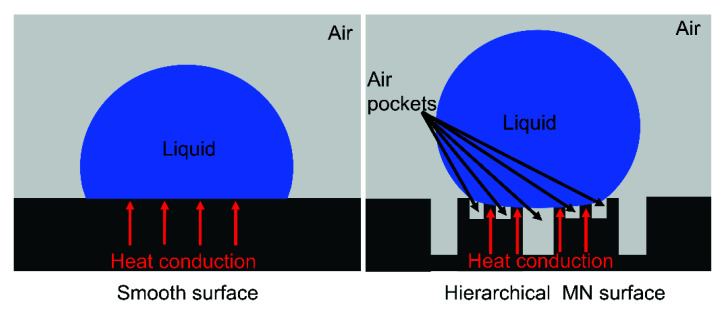
Heat conduction paths of smooth surface and hierarchical micro-/nanostructured surface.

## Data Availability

The raw data supporting the conclusion of this article will be made available by the authors, without undue reservation.

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
