# Peer review of "Bio-Inspired Hierarchical Micro-/Nanostructures for Anti-Icing Solely Fabricated by Metal-Assisted Chemical Etching"

_micromachines, 2022, doi:10.3390/mi13071077_

Round 1
Reviewer 1 Report
I sugguest to publish this manuscript after minor revised.
(1) Don't write the abbreviation "MN" in abstract. Reader will confuse it.
(2) Authors show this report is a cost-effective and scalable methodology for producing a hierarchical micro/nanostructured silicon surface solely by metal-assisted chemical etching in abstract. But they didn't summary a comparison table (comparison with literatures or tranditional method) to prove their concept.
Reviewer 2 Report
The article was interesting to read and it is written in understandable language. But there are a few notes and recommendations:
1) Line 23: the abbreviation МН was not introduced in the abstract and is incomprehensible at the first reading. I propose to remove the abbreviation here.
2) Line 40 introduces the abbreviation hMN, which is not used anywhere else, and then line 48 introduces the abbreviation MN. I propose to introduce only the abbreviation MN in line 40.
3) References 28 and 38 are strange.
4) There is not enough description of the equipment with which the samples were created and characterized. That’s why I disagree with the statement that this approach "is cost-effective, scalable, and excludes the use of vacuum environment and high-temperature processes". The authors use ultraviolet lithography, thermal evaporation, thermal oxidation at 1100 ℃ and deposition of amorphous fluorocarbon film. This statement should be replaced or explained in more detail in the text.
5) What explains the effect of a significant improvement in the anti-icing properties of MN structures compared to other structured surfaces? Where and how can this effect be applied? It needs to be discussed in the text.
Reviewer 3 Report
In this article, the superhydrophobic surface has been fabricated using a cost-effective and scalable approach. The authors have investigated and reported the results nicely and the results are also very interesting. I, therefore, recommend the publication of this article after addressing all my below comments.
1. 1. Authors have stated, “Most of these methods rely on costly equipment and require a high-temperature process or vacuum environment. Moreover, many of them require two different fabrication processes or materials to form the hierarchical micro/nanostructures”. To support these statements, appropriate references should be incorporated.
2. 2. In the experimental section, authors should first discuss from where they have obtained the chemicals and hierarchical MN surfaces before how they have utilized them to fabricate the superhydrophobic surfaces.
3. 3. The authors should examine the self-cleaning behavior of the proposed material. The obtained contact angle and sliding angles are undoubtedly good, but researchers have already fabricated several materials that demonstrate a contact angle of ~175° or higher. Hence, is 156° good enough to show outstanding self-cleaning behavior, as the authors claimed in the conclusion section?
4. 4. The introduction section seems to be very concise. More discussion is required about the recent progress in artificial superhydrophobic materials and their fabrication techniques etc. For this, the below articles could be of author’s interest:
a. https://doi.org/10.1080/03602559.2018.1447128
b. https://doi.org/10.1155/2013/486253
c. DOI: 10.1039/B412803F
d. https://doi.org/10.1016/j.molstruc.2019.127342
e. https://doi.org/10.1016/j.porgcoat.2019.105306
f. https://doi.org/10.1002/slct.202001092
g. https://doi.org/10.1007/s10971-022-05853-6
h. https://doi.org/10.1007/s11831-021-09689-1
i. https://doi.org/10.1557/jmr.2004.19.2.628
5. 5. There is a number of grammatical errors present in the manuscript; authors should identify them and correct them before publication. Such as at some places, authors have written “2876 s” and in some places, “2876s”. It should be identical throughout the manuscript. In some places, the degree symbol (°) is not correct.
Round 2
Reviewer 3 Report
The quality of the revised manuscript has been improved, I, therefore, recommend publishing this article.